# Classification of breast cancer histopathological images using interleaved DenseNet with SENet (IDSNet)

Xia Li[1], Xi Shen[1], Yongxia Zhou[1], Xiuhui Wang[1], Tie-Qiang Li[1,2,3]*

**1** College of Information Engineering, China Jiliang University, Hangzhou, China, **2** Department of Clinical Science, Karolinska Institutet, Intervention and Technology, Stockholm, Sweden, **3** Department of Medical Radiation and Nuclear Medicine, Karolinska University Hospital, Stockholm, Sweden

* tie-qiang.li@ki.se

**Data Availability Statement:** All relevant data are within the paper and its Supporting Information files.

## Abstract

In this study, we proposed a novel convolutional neural network (CNN) architecture for classification of benign and malignant breast cancer (BC) in histological images. To improve the delivery and use of feature information, we chose the DenseNet as the basic building block and interleaved it with the squeeze-and-excitation (SENet) module. We conducted extensive experiments with the proposed framework by using the public domain BreakHis dataset and demonstrated that the proposed framework can produce significantly improved accuracy in BC classification, compared with the state-of-the-art CNN methods reported in the literature.

## Introduction

Breast cancer (BC) is the most common type of cancer in women worldwide and has very high mortality. According to a recent report from the American Cancer Society, the number of new BC cases in the USA is about 268,600 in 2019. Early detection and diagnosis can make treatment more successful and help improve the BC survival rate [1]. Despite the extensive screening programs based on mammography worldwide, the histopathological image classification of BC is still very important for BC diagnosis and dictates the use of potentially curable therapy. However, diagnostic disagreements among different pathologists can be remarkably high, especially for the preinvasive lesions. Recently, Convolutional Neural Network (CNN) has attracted much attention for the analysis of histopathological images, because of its steadily improving performance that is nearly as accurate as or better than human experts [2]. Bayramoglu et al. [3] proposed a scalable and magnification independent CNN approach to improve the generalizability and speed of learning. Spanhol et al. [4] used pre-trained Alexnet to extract features and fed them into a task-specific classifier to improve CNN classification. They tested further a Deep Convolutional Activation Feature for Generic Visual Recognition (DeCAF) model and reported somewhat mixed results in comparison with the Alexnet approach [5]. Wei et al. [6] considered class and sub-class labels of breast cancer as prior knowledge and utilized GoogLeNet as their basis networks. Song et al. [7] used the VGG-VD model to extract

**Funding:** This work was supported by China Scholarship Council, Zhejiang Natural Science Foundation of China (No. LY18E070005) and the National Natural Science Foundation of China (No. 51377186).

**Competing interests:** The authors have declared that no competing interests exist.

image feature and designed a new adaptation layer for BC classification. Nahid et al. [8] combined CNN with a long short-term memory (LSTM) architecture for feature extraction and in the classification stage they used softmax and support vector machine (SVM) layers for decision-making of the BC categories. Araújo et al. [9] proposed recently a CNN architecture designed to retrieve information at different scales and achieved an accuracy of 83.3% for carcinoma/non-carcinoma classification and a sensitivity of 95.6% for BC cases.

To reduce the error rate in object recognition, Densely Connected Convolutional Networks (DenseNet) was introduced firstly by Huang et al. [10], in which each layer is directly connected to every layer in front of it. This network alleviates the vanishing-gradient problem, strengthens feature propagation, encourages feature reuse, and substantially reduces the number of parameters. To enhance the representational quality of a network by modeling channel-wise relationships in a computationally efficient manner, Hu et al. [11] proposed a Squeeze-and-Excitation Network (SENet) with lightweight gating mechanism, with this approach the network learns to use global information to selectively emphasize informative features and suppress less useful ones.

The Breast Cancer Histopathological Database (BreakHis) has been widely used in some of the CNN based classification studies [12]. We also used this dataset in the current study. The histopathological images of the BreakHis dataset have fine-grained appearances and are difficult to classify. In order to improve the classification accuracy, it is necessary to emphasize the details of the images and more local information. Therefore, we propose an architecture based on interleaved DenseNet with SENet (IDSNet) to tackle the problem. Since the DenseNet can enhance the feature delivery and SENet can boost effectiveness on feature selection, the proposed IDSNet does not only utilize the deeper information with higher complexity, but also merges the shallow information. Moreover, the IDSNet architecture uses global average pooling in the classification network to mitigate the lack of computing resources and network over-fitting caused by the large number of parameters.

In this study, we used the public domain BreakHis dataset to conduct experiments with the proposed IDSNet framework. The whole dataset were divided into training, validation and testing sets. As the standard method in the field of deep learning [13], the data augmentation method was used for the training set. In the following sections, we provide more experimental details and summarize the classification performance of the proposed framework in comparison with the other state-of-the-art methods reported in the literature [4, 5].

## Materials and methods

### The proposed IDSNet

DenseNet-121 was used in our proposed network architecture, in which each layer was directly connected to every other layer in a feed-forward fashion. As shown in Fig 1, it consists four dense blocks, three transition layers and a total of 121 layers (117-conv, 3-transition, and 1-classification). As described in the original DenseNet paper [10], each conv layer corresponds to a composite sequence of operations consisting of batch normalization (BN)-Relu-Conv. The Classification subnetwork includes $7 \times 7$ global average pooling, 1000D fully-connected layer, and softmax.

We fine-tuned the DenseNet-121 architecture to better fit the BreakHis dataset. The last pooling and linear layer following Dense Block 4 were removed and the feature-maps extracted from each transition layers and the output of Dense Block 4 were feed into their connected SENet module followed by the classification sub-networks for BC classification. The proposed IDSNet architecture is schematically shown in Fig 2.

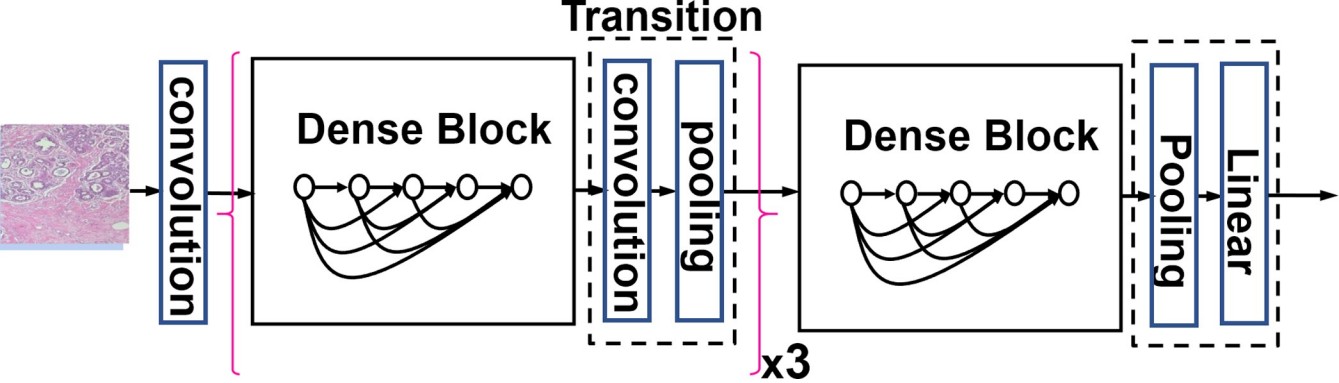

**Fig 1. The DenseNet-121 architecture.**

The initial convolution layer in this proposed CNN architecture corresponds to the sequence of a batch normalization (BN) [14] with a rectified linear unit (ReLu) [15] and a convolution (Conv). The Conv layer used a filter of $7 \times 7$ matrix with stride 2. The output from the convolution layer is used as input to the Dense Block of the DenseNet to further improve the information flow between layers. The transition layer consists of a convolution and a pooling layer. The output of the convolution goes through the SENet module to extract more channel information and is further classified in the classification sub-network. To extract more feature information, we stacked sequentially four copies of the basic building blocks consisted of Dense Blocks, transition layer, SENet module and classification sub-network. Outputs from each of the building block were concatenated and fed into the final fully connected layer. The detailed features of the proposed IDSNet framework will be discussed in the following sections.

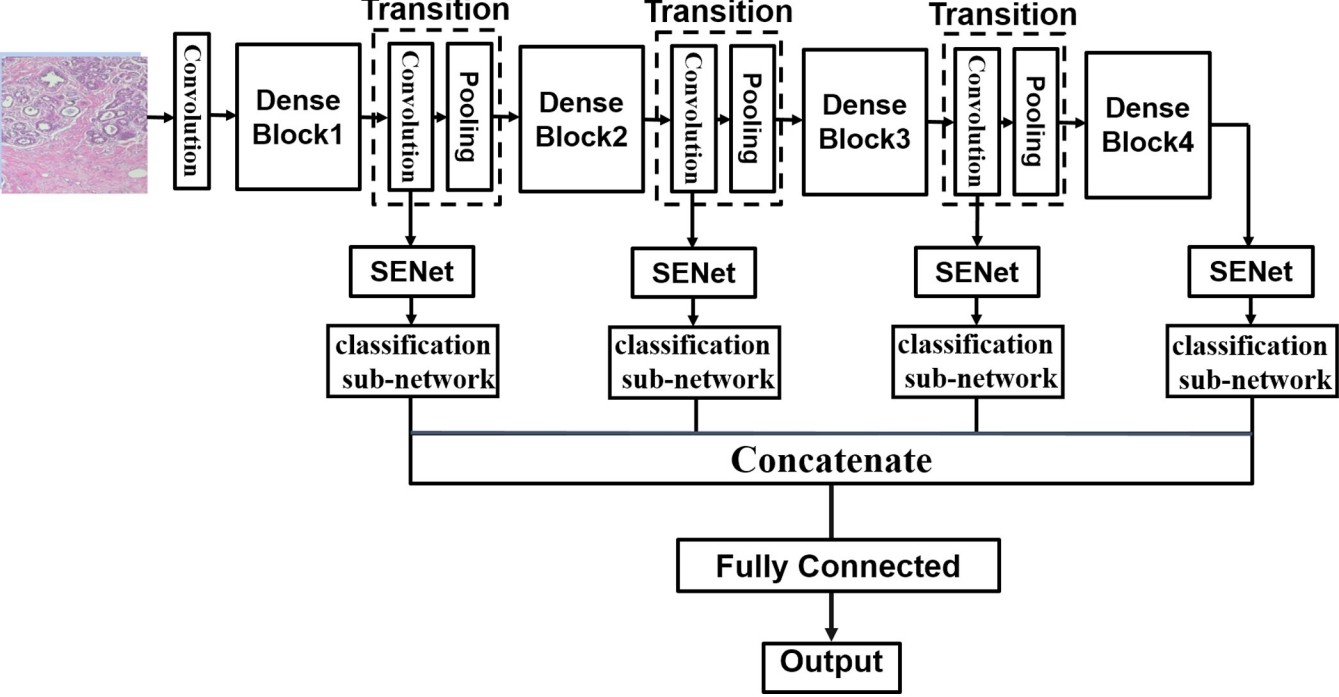

**Fig 2. Overview of the proposed IDSNet architecture.**

## The dense block

The Dense Block is an important part of the DenseNet for improving the information flow between layers. It is composed of BN, ReLu and $3 \times 3$ Conv. The specific formula is shown as follows,

$$x_1 = H_l([x_0, x_1, \ldots, x_{l-1}]) \tag{1}$$

where $[x_0, x_1, \ldots, x_{l-1}]$ refers to the concatenation of the feature-maps produced in layers 0,1,…,$l-1$, $H_l(\cdot)$ is defined as a composite function of three consecutive operations on the input of $l^{\text{th}}$ layer. The structure of Dense Block is shown in Fig 3.

**The transition layer.**   As illustrated in Fig 2, between two dense blocks, a transition layer is used for changing the size of the feature maps. It consists of a BN, a ReLu, a $1 \times 1$ conv and a $2 \times 2$ average pooling layer. Its specific structure is shown in Fig 4.

The convolution is responsible for the study of features by extracting features from the previous layer's output. The extracted features share a convolution kernel, also called a filter, which includes a set of weights. All local weight values need to be passed through an activation function (such as ReLU, sigmoid) to increase their nonlinearity. The convolution process can be expressed as,

$$\mathbf{z}^l = \mathbf{W}^l \cdot f_1(\mathbf{z}^{(l-1)}) + \mathbf{b}^l \tag{2}$$

where $\mathbf{z}^l$ is the $l^{\text{th}}$-layer neuron status, $f_l(\cdot)$ the activation function. $\mathbf{w}^l$ and $\mathbf{b}^l$ are the weight matrix and bias from $(l-1)^{\text{th}}$ to the $l^{\text{th}}$, respectively.

The pooling layer reduces the dimensionality of each feature map but retains the most important information. Max pooling and average pooling are used in our work. In case of max pooling, a spatial neighborhood (for example, a $2 \times 2$ window) is defined and the largest element is taken from the rectified feature map within that window. Instead of taking the largest element we take the average of all elements in that window in case of average pooling. In our experiments, we used a $3 \times 3$ window for max pooling and a $2 \times 2$ window for average pooling. The strides for both pooling layers are 2.

## The SENet architecture

In the proposed IDSNet architecture, the extracted feature-maps from DenseNet are fed into SENet modules for receiving more channel wise information. It uses global information to selectively emphasize informative features and suppress less useful ones. It introduces weights to each feature map in the layers. This is a composite function of five consecutive operations: a channel wise global average pooling [16], a fully connected (FC) layer, a ReLU, a fully connected layer followed by a sigmoid. The sigmoid activations function as channel weights adapted to the input-specific descriptor. There is a minor increase in terms of the number of parameters and computation loads because of the extra layers like FC and pooling operations. The unique structure of this "Squeeze-and-excitation" (SE) Net is illustrated in Fig 5 and it can be used with any standard architecture. SENet introduces intrinsically dynamics conditioned on the input to boost feature discriminability.

**The squeeze.**   The SENet achieves the squeezing operation by global average pooling to generate channel-wise statistics $\mathbf{Z} \in R^C$. The $k^{\text{th}}$ element of $\mathbf{Z}$, $z_k$, is calculated as

$$z_k = F_{sq}(u_k) = \frac{1}{H \times W} \sum_{i=1}^{H} \sum_{j=1}^{W} u_k(i, j), \tag{3}$$

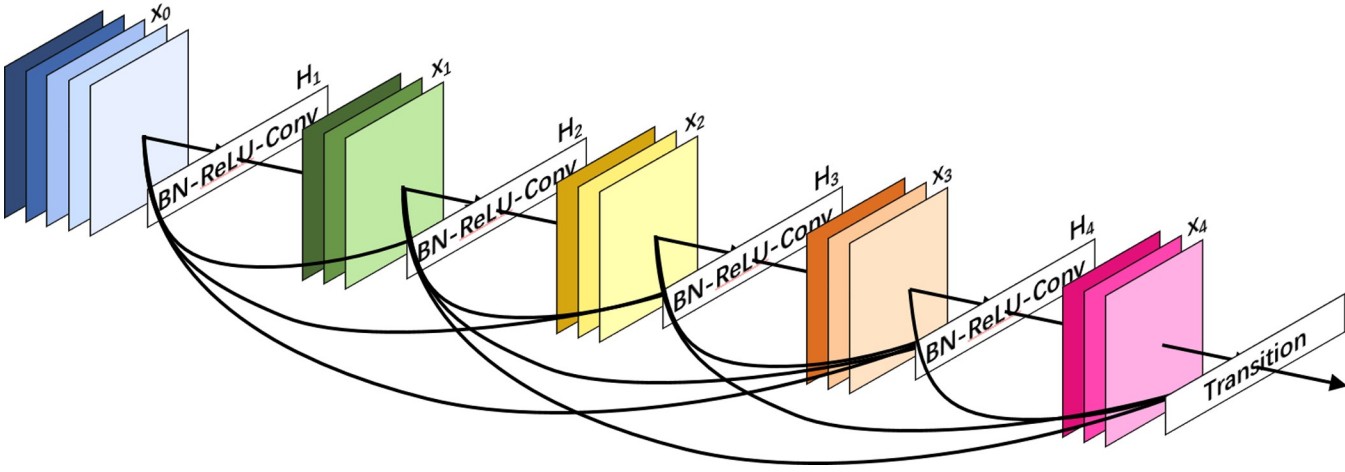

**Fig 3. The dense block.**

Where $F_{sq}(\cdot)$ is the operation of squeezing, $u_k$ is the $k^{th}$ feature map with spatial dimension H×W, and $k = 1,2,\ldots,C$.

**The excitation.** The excitation operation can help capture channel-wise dependencies and greatly reduce the number of parameters and calculations. Excitation mainly consists of two FC layers and two activation functions, which can be written as

$$\mathbf{S} = F_{ex}(\mathbf{Z}, \mathbf{W}) = \sigma(g(\mathbf{Z}, \mathbf{W})) = \sigma(\mathbf{W}_2\delta(\mathbf{W}_1\mathbf{Z})), \tag{4}$$

where $\mathbf{S} = \{s_1, s_2, \ldots, s_C\}$, $s_k \in R^{H\times W}(k = 1, 2, \ldots, C)$, $F_{ex}(\cdot)$ is the operation of excitation. $\mathbf{W}_1 \in R^{\frac{C}{r}\times C}$, $\mathbf{W}_2 \in R^{C\times\frac{C}{r}}$ and $r$ is a hyper-parameter ratio which can vary the capacity and computational cost. $\delta(x) = \max(0,x)$ refers to the ReLU for reducing the probability of the vanishing gradient [17] and $\sigma(x) = \frac{1}{(1+e^{-x})}$ sigmoid function. The final output $\tilde{x}_k(k = 1, 2, \ldots, C)$, is obtained by multiplying the input channels with their respective weights,

$$\tilde{x}_k = F_{scale}(u_k, s_k) = u_k \cdot s_k, \tag{5}$$

Where $\tilde{x}_k \in R^{H\times W}$, $F_{scale}(u_k, s_k)$ refers to channel-wise multiplication between the scalar $s_k$ and the feature map $u_k$.

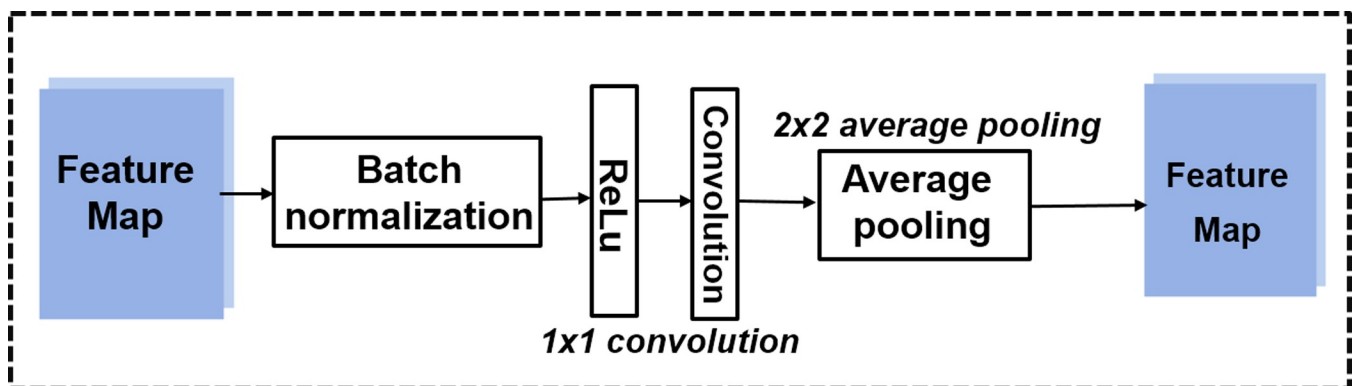

**Fig 4. The transition layer.**

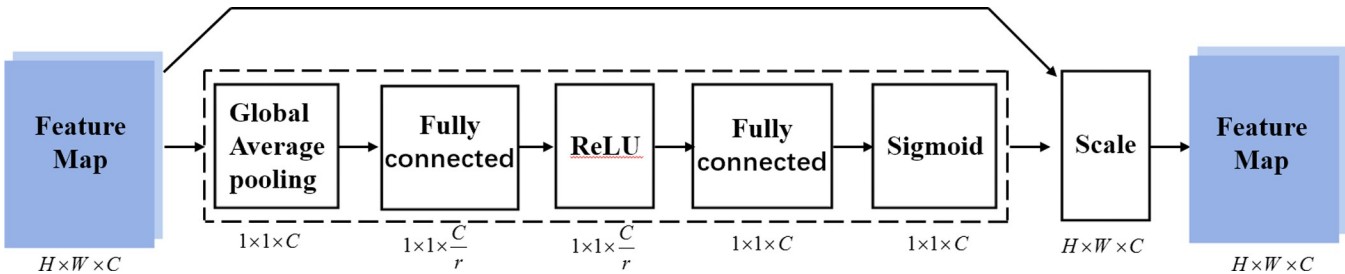

**Fig 5. The SENet architecture.**

### The classification sub-network

The classification sub-network is used to reduce parameters and distinguish the categories of BC. As shown in Fig 6, it consists of a global average pooling, a BN and a softmax function.

**The global average pooling.** The Global average pooling sums out the spatial information, so it is more robust for the input spatial translation. Feature maps shrink to a statistic and over fitting is avoided at this layer. The following BN is used to speed up the training process and make training more stable.

**The softmax classifier.** The softmax function takes an input vector K of real number and normalizes the outputs so that they sum to 1 and can be directly treated as probabilities proportional to the exponentials of the input numbers. The standard (unit) softmax function $\sigma$ : $R^K \to R^K$ is defined by the formula:

$$\sigma(\mathbf{z})_j = \frac{e^{z_j}}{\sum_{k=1}^{K} e^{z_k}} j = 1, \ldots, K \; \mathbf{z} = (z_1, \ldots, z_K) \in R^K. \tag{6}$$

**The loss function.** We use the cross-entropy as the loss function. Cross-entropy loss measures the performance of a classification model whose output is a probability value between 0 and 1. It increases as the predicted probability diverges from the actual label and can be calculated in binary classification,

$$loss = -(y \log(p) + (1 - y)\log(1 - p)) \tag{7}$$

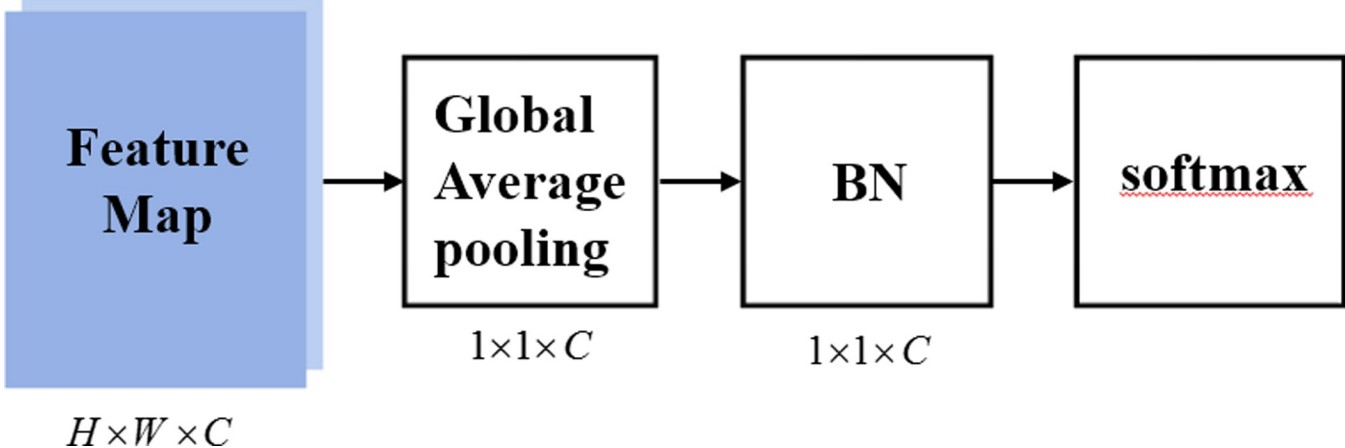

**Fig 6. The classification sub-network.**

where *y* is the correct classification and *p* is the predicted probability. We aim to reduce the loss value through continuous learning of the network.

## The BreaKHis dataset

In this study, we used the images from the BreakHis dataset [12], which contains 7909 images from 82 patients with four different magnifications (40×, 100×, 200×, 400×). The dataset was collected at Pathological Anatomy and Cytopathology (P&D) Lab, Brazil. It contains microscopic biopsy images of benign and malignant breast tumors. Slides of biopsy specimens for breast tissue were stained with hematoxylin and eosin (H&E). To cover the whole region of interest (ROI) identified by pathologist, samples were captured using the lowest magnification factor (40×) and the interested areas from initial ROI were manually magnified to a higher factor (100×). This process was repeated for 200× and 400× magnifications. Images are in 3-channel RGB (Red-Green-Blue), 8-bit depth in each channel, PNG (Portable Network Graphics) format without compression and dimension of 700 × 460 pixels. A typical set of images from the BreaKHis dataset are shown in Fig 7. The details of the dataset in terms of the magnification and lesion types are summarized in Table 1. In the study, the BC images were adjusted to 224 × 224 pixels and normalized before being fed as input to the convolution layer and used to extract feature maps.

## Training & testing protocol

We used a workstation based on Ubuntu 16.04.3 LTS system and a NVidia GeForce GTX Titan X 12GB GPU to carry out the study. Python and TensorFlow frameworks were the development environments. The network implementation was based on the Adam optimization algorithm, which is known to be fast and effective for computer vision related problems.

In this work, we combined transfer learning with fine-tuning method. We utilized the public domain dataset ImageNet [18] to pre-train DenseNet121 and used the obtained weight values as initialization weight values for our experiments. We used 50% of the patient data in a randomized fashion for training, 20% for validating and 30% for testing. The division of the dataset was not subject or magnification specific. In other words, the images of different magnification factors from the different patients were mixed in the process of constructing the datasets for training, validating and testing. There was no overlap among these three sets of imaging data. The training set was used to train the model and optimize the connection parameters for the different neurons. The validation set was used to select the model, while the test set was only used to test classification accuracy and model reliability. In order to reduce the contingency of the experiment, we repeated the experiment for three times with randomized inputs each time.

Besides the default configuration parameters, other adjusted hyperparameters for the network training included the followings: 60 epochs with a batch size of 64, momentum = 0.9, transformation rate = 0.05, training sample size = 3954, validation sample size = 2370, the growth rate of the DenseNet $k$ = 12, and reduction rate $r$ = 8. We adopted different values of learning rate into five stages. The initial learning rate was $3\times10^{-3}$ to accelerate the training of the network. The learning rate was lowered by a factor of 2 at epochs 25 to 30. To avoid missing the best point, the learning rate was set up to $3\times10^{-5}$ at epoch 35 and $1\times10^{-5}$ at epoch 40.

In order to improve the performance of classification, prevent over-fitting and enhance the robustness of the network, fine-tuning with data augmentation method was used for the training dataset. We adopted rotation (90°/180°/270°) and flipping (horizontal mirror/vertical mirror) to expand the original training data by 5 times.

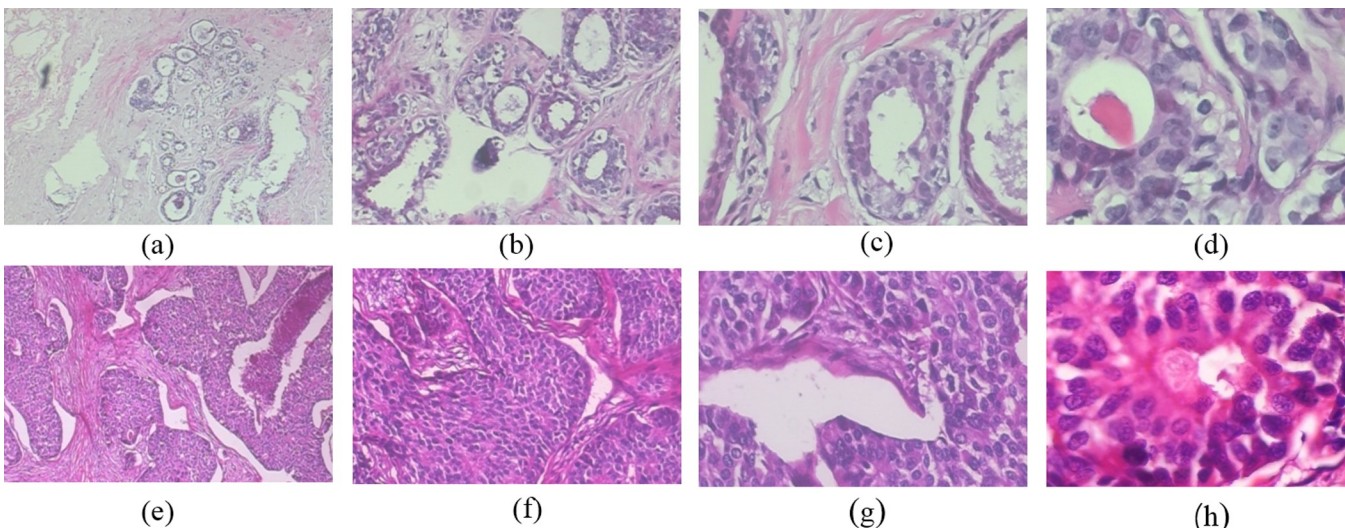

**Fig 7. Image samples from BreakHis dataset.** (a)~(d) benign tumor and (e)~(f) malignant tumor with the magnification factor of 40×, 100×, 200×, 400×.

**The evaluation metrics.** There are two ways to evaluate the effectiveness of machine learning on medical data sets [12]. In our experiment, the accuracy of different magnification factors is calculated. Firstly, we evaluated the accuracy on patient level and assessed the patient recognition rate (PRR). Assume the number of cancer images of patient $N_p$, $N_{rec}$ is the cancer images classified correctly and $N$ the number of total patients, then the PRR is defined as

$$PRR = \frac{\sum patient\ score}{N},\tag{8}$$

in which $Patient\ score = \frac{N_{rec}}{N_p}$.

We also evaluated the recognition rate at the image level to provide assessment for the image classification accuracy. Let $N_{all}$ be the number of cancer images in the testing set, $N_r$ the images correctly classified, then the image recognition rate (IRR) is

$$IRR = \frac{N_r}{N_{all}}\tag{9}$$

To assess the effectiveness of the DenseNet for BC classification, we compared the performances of the proposed DenseNet with VGG16 [19] and Resnet50 [20] networks using the BreakHis dataset. Furthermore, to evaluate the performance of SENet and classification subnetwork, we conducted the following three cases of experiments: 1) the complete IDSNet

**Table 1. Image distribution by magnification factor and class.**

| Magnification | Benign | Malignant | Total |
|---|---|---|---|
| 40× | 625 | 1370 | 1995 |
| 100× | 644 | 1437 | 2081 |
| 200× | 623 | 1390 | 2013 |
| 400× | 588 | 1232 | 1820 |
| Total | 2480 | 5429 | 7909 |
| Patients | 24 | 58 | 82 |

based on DenseNet interleaved with SENet and classification sub-network; 2) DenseNet combined with classification sub-network only; 3) DenseNet combined with SENet module only.

## Results

Table 2 shows the experimental results of three previously reported CNN networks: DenseNet, VGG16 and Resnet50. These networks are potential building stones for further development of the study.

As shown, the DenseNet-121 can achieve more accurate BC classification than the other two networks. The PRR metric for the DenseNet-121 is improved by about 2–8% in comparison with VGG16 and Resnet50, while the improvement in IRR metric for the DenseNet-121 is varied from 2–9% depending on the magnification factor of the images.

The results for the three possible types of combination cases among DenseNet, SENet and classification sub-network are summarized in Table 3. Compared to the results shown in Table 2, it is clear that the combination of DenseNet with SENet module or classification sub-network can improve significantly the BC classification accuracy for all the histological images irrespective of the magnification factors. The proposed IDSNet has the best performance in terms of both PRR and IRR metrics for almost all magnifications except for IRR at the zoom factor of 100×. Overall, we observe that the IDSNet can improve further the classification accuracy by about 2–3% compared with the other combinations. The best PRR result showed at least 4% increase in accuracy for the zoom factor of 100×, while the best IRR result exhibited at least 0.5% increase in accuracy for the zoom factor of 400×.

We compared also the performance of proposed approach with other state-of-the-art methods reported in the literature [4, 5]. The results based on these methods are shown in Table 4. It is apparent that the proposed IDSNet can improve BC recognition rate for images of all magnification factors at patient level by about 1–7% in comparison with the reported literature results. The least improvement is 0.9% whereas the best improvement is 6.7%. At image level, the improvement appears to be less obvious, particularly at the worst case of low magnification of 40×, the classification accuracy is even 0.5% lower.

## Discussion

In this study, we proposed a novel CNN architecture IDSNet for BC classification in histological images of different magnifications. To improve the delivery and usage of feature information, we constructed the model by sequentially stacking four copies of the building blocks consisted of the basic DenseNet, SENet module and the classification sub-network. Furthermore, we aggregate all information from the different levels of depth by concatenating the feature maps extracted from the four copies of sequentially stacked building blocks. An additional classification sub-network was utilized to perform the binary classification task. We

**Table 2. Performance of BC classification using VGG16, Resnet50 and DenseNet-121.** The best performance is highlighted by boldface.

| Metric | Network | Magnification factor | | | |
|---|---|---|---|---|---|
| | | 40× | 100× | 200× | 400× |
| PRR | VGG16 | 74.9 | 76.0 | 78.7 | 79.0 |
| | Resnet50 | 78.1 | 78.2 | 77.0 | 78.2 |
| | DenseNet-121 | **82.2** | **83.9** | **80.6** | **82.6** |
| IRR | VGG16 | 72.5 | 77.7 | 77.2 | 77.5 |
| | Resnet50 | 72.5 | 76.3 | 75.0 | 80.4 |
| | DenseNet-121 | **81.8** | **79.3** | **81.4** | **83.2** |

**Table 3. Performance of BC classification by the different network combinations: Case1) the complete IDSNet based on DenseNet interleaved with SENet and classification sub-network; Case2) the DenseNet combined with classification sub-network only; Case3) DenseNet combined with SENet module only.** The best performance is highlighted by boldface.

| Metric | Model | Magnification factor | | | |
|--------|-------|------|------|------|------|
| | | 40× | 100× | 200× | 400× |
| PRR | case 1 | **89.5±2.0** | **87.5±2.9** | **90.0±5.3** | 84.6±2.1 |
| | case 2 | 88.9±2.4 | 82.3±5.1 | 88.0±5.6 | 84.0±2.9 |
| | case 3 | 86.9±1.9 | 83.5±4.19 | 87.1±2.0 | **84.7±2.0** |
| IRR | case 1 | **89.1±3.6** | 85.0±5.1 | **87.0±6.0** | **84.5±3.6** |
| | case 2 | 88.7±3.7 | **85.4±2.8** | 86.9±5.8 | 83.2±4.9 |
| | case 3 | 87.1±1.91 | 81.9±6.9 | 84.4±6.3 | 84.0±3.9 |

conducted extensive experiments with the proposed framework by using the publicly available BreakHis dataset and demonstrated that the proposed framework can produce significantly improved BC classification performance, compared with the state-of-art CNN methods reported in the literature. The key finding is that appropriate integration of DenseNet, SENet, and classification sub-network is a feasible approach to attain robustness in BC histological image classification.

## The selection of the reduction ratio

The reduction ratio $r$ is a hyperparameter to regulate the capacity and complexity of the SE blocks in the SENet. With the increase of $r$ value the number of parameters in the network will be reduced, which can lead to undesirable results. However, if the value of $r$ is too small, the overfit of network may occur. Therefore, there is a trade-off between the performance and complexity. After systematic experimentations for a range of different $r$ values 2–20, we found that it is robust to set $r = 8$ for our study. However, using the same reduction ratio throughout a network may not be optimal because of the distinct functions of the different layers. In practice, further improvements may be achievable by tuning the $r$ value adaptively according to its functional role in each layer.

## Novelty of the architecture, pre-training and fine-tuning

When CNN networks go deeper from 10s to 1000s layers, the path from the input to the output layer becomes so large that the input information can vanish before it can reach the other end. DenseNets mitigate this problem by ensuring maximum information flow and connecting every layer directly with each other. In other words, DenseNets exploit the potential of the network through feature re-use instead of extracting the representation power from extremely

**Table 4. Performance of BC classification for the proposed IDSNet in comparison with other literature methods.** The best performance is highlighted by boldface.

| Metrics | Methods | Magnification | | | |
|---------|---------|------|------|------|------|
| | | 40× | 100× | 200× | 400× |
| PRR | AlexNet based [4] | 88.6±5.6 | 84.5±2.4 | 83.3±3.8 | 81.7±4.9 |
| | DeCaF CNN [5] | 84.0±6.9 | 83.9±5.9 | 86.3±3.5 | 82.1±2.4 |
| | IDSNet (this study) | **89.5±2.0** | **87.5±2.9** | **90.0±5.3** | **84.6±2.1** |
| IRR | AlexNet based [4] | **89.6±6.5** | 85.0±4.8 | 84.0±3.2 | 80.8±3.1 |
| | DeCaF CNN [5] | 84.6±2.9 | 84.8±4.2 | 84.2±1.7 | 81.6±3.7 |
| | IDSNet (this study) | 89.1±3.6 | **85.0±5.1** | **87.0±6.0** | **84.5±3.6** |

**Table 5. Parameter and model size for the different CNN models used to test the BreakHis dataset.**

| Network | VGG16 | Resnet50 | DenseNet-121 | IDSNet |
|---|---|---|---|---|
| Parameters | 14,765K | 23,788K | 7,138K | 7,664K |
| Model size | 169.1Mb | 272.6Mb | 82.6Mb | 88.65Mb |

deep or wide architectures. On the other hand, SENet can enhance effectiveness for feature selection. The proposed IDSNet does not only utilize the deeper information with higher complexity, but also merges the shallow information. Since the shallow layers contain a lot of detailed and local information, while the deep network layers contain rich semantic information [21]. Therefore, with the IDSNet framework we can efficiently extract the information in different dense blocks and send them to SENet module for obtaining more spatial information and weighting. The features from both the shallow and deep layers are aggregated by concatenation and sent to the final classification sub-network to improve the classification performance of the model.

The IDSNet architecture uses global average pooling in the classification sub-network to mitigate the lack of computing resources and network over-fitting associated with the large number of parameters and inflated model size. Excessive parameters will make the network's structure very complicated and consume computing resources. The use of the global average pooling can lead to better integration of the spatial information and shrink of the number of parameters. For better comparison, we summarized the number of parameters for the relevant network structures in Table 5.

Compared with VGG16 and Resnet50, the IDSNet achieved significantly improved performance with substantially fewer parameters and smaller model size. Compared to the DenseNet, our model has slightly more parameters and larger model size, which is quite acceptable for the achieved improvement in BC classification.

Transfer learning is an effective way to solve the problem with a relatively small training sample and use the pre-trained model to improve the training efficiency and generalization ability of the network. If the weights in a CNN network were not pre-trained through large datasets, the initial weight values would have to be set in a random fashion and the network convergence would be slow. In this study, we used transfer learning to select the initialization weights and adjusted their values through fine-tuning on basis of the actual BC histological images. In this way, we can accelerate the network convergence for model training.

## Conclusions

In this study, we configured a novel CNN architecture IDSNet by stacking multiple copies of the basic building unit consisting of the pre-trained Dense block, SENet and classification sub-network to extract more channel features and enhance the use of more important local information in BC biopsy specimens with fine grain features. The experimental results obtained from the BreakHis dataset have demonstrated that the framework can significantly improve the BC classification accuracy without notable expansion of the model. In future studies, we shall further explore the potential of IDSNet for classification of other types of medical images.

## Supporting information

**S1 File.**
(ZIP)

## Author Contributions

**Formal analysis:** Xi Shen.

**Funding acquisition:** Xia Li.

**Investigation:** Xiuhui Wang, Tie-Qiang Li.

**Methodology:** Xi Shen.

**Software:** Yongxia Zhou.

**Supervision:** Xiuhui Wang.

**Writing – original draft:** Xia Li.

**Writing – review & editing:** Tie-Qiang Li.

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
