## [Decision Letter · Decision Letter 0]

26 Feb 2020

PONE-D-20-01196

Classification of Breast Cancer Histopathological Images Using Interleaved DenseNet with SENet (IDSNet)

PLOS ONE

Dear Professor of MRI Physics Tieqiang,

Thank you for submitting your manuscript to PLOS ONE. After careful consideration, we feel that it has merit but does not fully meet PLOS ONE’s publication criteria as it currently stands. Therefore, we invite you to submit a revised version of the manuscript that addresses the points raised during the review process.

Please revised the paper by considering the reviewers' comments.

We would appreciate receiving your revised manuscript by Apr 11 2020 11:59PM. To enhance the reproducibility of your results, we recommend that if applicable you deposit your laboratory protocols in protocols.io, where a protocol can be assigned its own identifier (DOI) such that it can be cited independently in the future. For instructions see: http://journals.plos.org/plosone/s/submission-guidelines#loc-laboratory-protocols

We look forward to receiving your revised manuscript.

Kind regards,

Jie Zhang

Academic Editor

PLOS ONE

Journal Requirements:

"This work was supported by China Scholarship 460 Council, Zhejiang Natural Science Foundation of China (No. LY18E070005) and the National Natural Science Foundation of China (No. 51377186)."

Reviewers' comments:

Reviewer's Responses to Questions

**Comments to the Author**

1. Is the manuscript technically sound, and do the data support the conclusions?

Reviewer #1: Yes

2. Has the statistical analysis been performed appropriately and rigorously? 

Reviewer #1: Yes

3. Have the authors made all data underlying the findings in their manuscript fully available?

Reviewer #1: Yes

4. Is the manuscript presented in an intelligible fashion and written in standard English?

Reviewer #1: Yes

5. Review Comments to the Author

Reviewer #1: This paper proposes a novel convolutional neural network architecture by combining DenseNet and squeeze-and-excitation module. Experiments on the public domain BreakHis dataset demonstrate the strength of the proposed network with superior performance over the state-of-the-art methods. The idea is good and the experimental results are impressive. However, the paper can be further improved by the following aspects:

1. Fig.1 contains the classification sub-network, which contains pooling, BN and softmax function. Then, is the class probabilities obtained by the sub-network will be fed into a fully connected layer? This part is not clear. Please explain it.

2. In experiments, the experimental setting is not clear, is the database are divided based on patients? In other words, is the training set and testing set containing the samples from one patient? Additionally, does the training set contain samples with different magnification factors when training one model?

3. Please seriously check the grammar errors and typos, some examples are:

1) state-of-art -> state-of-the-art

2) line 82: used also -> also used

3) line 182: very minute increase, please check this collocation.

4) line 323: Table II shows the experimental results for three previously… -> Table II shows experimental results of three previously …

5) line 334: varied -> is varied

6. PLOS authors have the option to publish the peer review history of their article (what does this mean?). If published, this will include your full peer review and any attached files.

Reviewer #1: No

---

## [Author Response · Author response to Decision Letter 0]

6 Apr 2020

Dear Editor, 

Thank you for giving us the opportunity to submit a revised draft of the manuscript referred above. We appreciate the time and effort that you and the reviewer have dedicated to providing us valuable feedback on the manuscript. 

We have formatted the title page and text body according to the style required for PLOS ONE. We removed the Acknowledgements section in the manuscript and provided funding information in the Funding Statement section of the online submission form.

We are grateful to the reviewer for the insightful comments on the paper. We have been able to incorporate changes to reflect the suggestions provided by the editor and reviewer. We have highlighted the changes in the manuscript with tracking and provided also a clean copy of the manuscript. The point-by-point response to the reviewer’s comments and concerns are provided in “Response to the Reviewer”.

There is nothing to disclose except for the information provided in the Funding Statement section.

We look forward to hearing from you in due time regarding our submission and to respond to any further questions and comments you may have.

---

## [Editor Report · Decision Letter 1]

8 Apr 2020

Classification of Breast Cancer Histopathological Images Using Interleaved DenseNet with SENet (IDSNet)

PONE-D-20-01196R1

Dear Dr. Li,

We are pleased to inform you that your manuscript has been judged scientifically suitable for publication and will be formally accepted for publication once it complies with all outstanding technical requirements.

With kind regards,

Jie Zhang

Academic Editor

PLOS ONE

Additional Editor Comments (optional):

The authors have adequately addressed the reviewer's comments and the revised manuscript can be accepted.
---

## [Editor Report · Acceptance letter]

22 Apr 2020

PONE-D-20-01196R1 

Classification of Breast Cancer Histopathological Images Using Interleaved DenseNet with SENet (IDSNet) 

Dear Dr. Li:

I am pleased to inform you that your manuscript has been deemed suitable for publication in PLOS ONE. Congratulations! Your manuscript is now with our production department. 

With kind regards,

on behalf of

Dr. Jie Zhang 

Academic Editor

PLOS ONE